# New and Efficient Bioactive Glass Compositions for Controlling Endodontic Pathogens

**DOI:** 10.3390/nano12091577

**Published:** 2022-05-06

**Authors:** Bruna L. Correia, Ana T. P. C. Gomes, Rita Noites, José M. F. Ferreira, Ana S. Duarte

**Affiliations:** 1Department of Materials and Ceramic Engineering, CICECO-Aveiro Institute of Materials, Campus Santiago, University of Aveiro, 3810-193 Aveiro, Portugal; brunacorreia@ua.pt (B.L.C.); jmf@ua.pt (J.M.F.F.); 2Universidade Católica Portuguesa, Faculdade de Medicina Dentária, Centro de Investigação Interdisciplinar em Saúde, 3504-505 Viseu, Portugal; apgomes@ucp.pt (A.T.P.C.G.); rita.noites@ucp.pt (R.N.)

**Keywords:** bioactive glass, root canal treatment, reinfection, *Candida albicans*, *Enterococcus faecalis*, intracanal medication

## Abstract

Endodontic treatment aims to conserve teeth through removing infected tissue, disinfecting, and filling/sealing the root canal. One of the most important treatment steps is the removal of microorganisms to avoid reinfection and consequent tooth loss. Due to increased resistance to intracanal medications, new alternative procedures are needed. Thus, an intracanal medication is suggested using three bioactive glass (BG) compositions (BG1, BG2, and BG3) produced by the sol–gel method, with different molar contents of bactericidal oxides. The BGs were morphologically and physically characterized. Their ability to inhibit the growth of two oral pathogens responsible for the failure of endodontic treatments (*E. faecalis* and *C. albicans*) was also studied. The results suggest that BG2 and BG3 can inhibit the growth of *E. faecalis* after 48 h of incubation, and all BG samples have a significant effect on *C. albicans* survival.

## 1. Introduction

Tooth decay is one of the greatest causes of tooth loss in the world. This not only affects the patient’s quality of life, but also carries an economic burden due to the need for multiple reinterventions [1]. Caries, fractures, or restorative failure may result in extreme and deep lesions, allowing microorganisms to enter the pulp chamber. Inflammation or infection of the pulp or periapical tissues may occur, which may lead to necrosis, abscess, and eventually tooth loss [2,3,4].

Root canals are a dental procedure aimed at saving natural teeth. Also known as endodontic therapy, root canals consist of careful cleaning and chemical–mechanical disinfection, followed by the application of intracanal medication for disinfection and pain relief; lastly, the tooth is filled/sealed [2,3,4,5,6]. Although cleaning and shaping of the root canal space reduces microbiota, this first step of the clinical procedure is not completely efficient at removing all the microorganisms from the apical deltas, isthmi, and accessory root canals [7]. Thus, intracanal medication between appointments is recommended to further reduce bacteria and fungi in the root canal system, which means multiple visits may be required [8,9]. Nevertheless, some resistant microorganisms persist and pose a risk of onset and maintenance of periapical diseases.

Endodontic infections are polymicrobial, but two of the most frequently implicated microorganisms in endodontic infection and endodontic treatment failure are *Enterococcus*
*faecalis* and *Candida*
*albicans* [10]. *E. faecalis*, a gram-positive and facultatively anaerobic bacterium, is difficult to remove from the root canal due to its ability to invade areas that are difficult to access with mechanical instruments. It can also form biofilms that make it more resistant to antimicrobial agents and grow in environments that lack nutrients [11,12,13,14]. Furthermore, *E. faecalis* is resistant to alkaline pH, essentially due to the presence of proton pumps that lower the pH in such conditions [11,12,13,14]. *C. albicans* is a dimorphic fungus (switching from yeast to hypha) and is the most common isolate of persistent endodontic infections. *Candida albicans* strains integrate polymicrobial biofilms, which increase their resistance to antifungal agents [15,16]. The ability of *C. albicans* to survive in environments with pH < 2 and pH > 10 is also related to the presence of pump mechanisms, such as P-type ATPase pumps that allow it to regulate the pH of the cytosol [17].

Calcium hydroxide (Ca(OH)_2_) has been the most widely used intracanal medication since the 1920s [18]. Its antimicrobial properties are related to its high pH (12.5 to 12.8). The alkalinizing effect of Ca(OH)_2_ results from the release and diffusion of calcium and hydroxyl ions, which can damage the protective barriers of microorganisms, such as lipopolysaccharides (LPS) [18,19]. Although exerting antimicrobial activity, it has been demonstrated that Ca(OH)_2_ has become ineffective in the inactivation of microorganisms such as *E. faecalis* and *Candida* species [18,19]. Moreover, *in vitro* studies demonstrated that the use of Ca(OH)_2_ as a long-term intracanal medication reduces the microhardness of the dentin and consequently the resistance of the tooth to fractures. Clinically, 32–40% of root fractures are associated with the use of Ca(OH)_2_ in treatment [20,21]. It has also been proven that the remaining particles (which are not completely removed during chemo–mechanical instrumentation and disinfection) affect the sealant’s bond with the dentin, which increases the probability of treatment failure [20,22].

To overcome the drawbacks related to the use of Ca(OH)_2_ as intracanal medicament, new materials such as bioactive glasses (BGs) have been investigated. A good alternative to Ca(OH)_2_ should promote dentin remineralization and bone regeneration by the formation of a hydroxyapatite layer [20]. Similar to Ca(OH)_2_, which is used in powder form for endodontic applications, BGs also need a vehicle (such as distilled water or saline solution) that facilitates accurate placement using a syringe. At the same time, the vehicle must support ion dissociation, solubility, and reabsorption [23].

Since the discovery of the first bioactive glass in the late 1960s by Larry Hench et al., BGs have been increasingly used in tissue regeneration and tissue engineering [24,25]. BGs have been demonstrating their ability to stimulate osteocunduction, osteoinduction, and biocompatibility in *in vivo* and *ex vivo* studies [24]. Extremely relevant in root canal treatment, BGs may have combined antimicrobial properties, usually associated with the release of certain ions and alkalinization of the medium [20]. In addition to their osteo-regenerative effects, antiseptic activity has been reported for many promising BGs, including BGs doped with Zn-, Cu-, and Sr- ions [26,27,28]. Antimicrobial activity by metal-doped BGs has been demonstrated against different bacteria: *Escherichia coli*, *Staphylococcus aureus*, *Aggregatibacter actinomycetemcomitans*, and *Porphyromonas gingivalis*; however, to the best of our knowledge, only Bioactive Glass S53P4 inhibits both of the key endodontic pathogens: *C. albicans* and *E. faecalis* [27,29,30].

BGs can be prepared by melting route or by sol–gel. The relevance of the sol–gel method has been increasing because: (i) it is more versatile in terms of variety of compositions that can be prepared; (ii) it allows using high-purity chemical precursors and close control of the composition; (iii) the synthesis process is based on chemistry and involves dissolution of precursors to obtain a homogeneous and transparent solution (sol), followed by a crosslinking copolymerisation reaction that occurs at room temperature [31]. Basically, the sol–gel method consists of using organic and inorganic precursors that are submitted to different processes (dissolution, hydrolysis, condensation) to form a homogeneous solution (sol). With condensation, the viscosity of the sol gradually increases and ends up forming a gel, which is then dried and finally calcined to obtain an amorphous, glassy, porous structure [24]. A sol–gel-derived BG composition (60 SiO_2_—34 CaO—4 MgO—2 P_2_O_5_ system (mol%)) has been revealed to be suitable for biological application and for 3D printing of porous scaffolds for bone regeneration and tissue engineering applications [32].

Inspired by that BG composition [32], the present study aims to design, synthesize through sol–gel, and characterize new BG powders endowed with antimicrobial activity. This will be accomplished while decreasing the content of the main network former (SiO_2_) to 50 mol.%, while incorporating distinct molar contents of metal oxides (ZnO, CuO, and SrO) known for their bactericidal roles. The lower degree of glass-network connectivity (silicon–oxygen backbone) is expected to contribute to the release and supply of the relevant therapeutic Zn^2+^, Cu^2+^, and Sr^2+^ ions from the BGs to improve their antibacterial and antifungal potential, especially against endodontic pathogens.

## 2. Materials and Methods

### 2.1. Preparation of the Bioactive Glass Samples

The synthesis of the BG samples with different molar compositions displayed in Table 1 was performed by the sol–gel method, following the procedure described elsewhere [33]. The quantities of the reagents for each sample were calculated in order to obtain a total of 0.2 mol BG per batch. Briefly, two solutions were prepared simultaneously, under magnetic agitation at 600 rpm for 30 min. Solution A, containing the precursor network formers, consisted of 20 mL of distilled water to which 2 drops of nitric acid (HNO_3_; Labkem, Barcelona, Spain), 29.08 g of tetraethyl orthosilicate (TEOS, Si(OC_2_H_5_)_4_; Sigma-Aldrich, Darmstadt, Germany), and 2.91 g of triethyl phosphate (TEP, (C_2_H_5_)_3_PO_4_; Sigma-Aldrich, Darmstadt, Germany) were added. Solution B, containing the network modifiers, was composed of 20 mL of distilled water to which 2 drops of nitric acid (HNO_3_; Labkem, Barcelona, Spain) and the required amounts of nitrate precursors (Ca(NO_3_)_2_ · 4 H_2_O, Mg(NO_3_)_2_ · 6 H_2_O, Sr(NO₃)₂, Cu(NO_3_)_2_ · 3 H_2_O, and Zn(NO_3_)_2_ · 6 H_2_O, all from Sigma-Aldrich, Darmstadt, Germany) were added, depending on the specific composition desired. After stirring for 30 min, both transparent solutions were mixed under magnetic stirring for 1 h at 600 rpm. Then, the obtained transparent and homogeneous overall solution (sol) was poured into Petri dishes and placed in an oven for 24 h at 100 °C. After 24 h, the solidified gel was crushed into a fine powder using a mortar and pestle and then subjected to heat treatment.

### 2.2. Thermal Treatment and Milling

The bioactive glass samples BG1, BG2, and BG3 were calcined at 700 °C for 2 h with a heating rate of 5 °C min^−1^.

The milling process was carried out according to the procedure described by Ben Arfa et al. [34,35]. The calcined powders were placed in a 300 cm^3^ alumina jar with 10 mm diameter zirconia spheres at a powder-to-sphere ratio of 1:10. Acetone was also added at a powder-to-acetone ratio of 1:1. Finally, the alumina jar was placed in a planetary mill (Ceramic instruments, Sassuolo, Italy, type S2-1000) at 390 rpm for 1 h. The powder was placed in glass flasks inside an oven at 100 °C for 24 h for drying and to ensure that there was no contamination for future biological tests.

### 2.3. Physical and Morphological Characterization of the BG Samples

To evaluate thermal transformation and weight change of the BG powder samples with temperature changes, differential thermal analysis (DTA) and thermogravimetric analysis (TG) were performed using a simultaneous thermogravimetric analyzer (STA - NEXTA STA300, Hitachi high-tech, Tokyo, Japan) using a heating rate of 5 °C min^−1^ to a maximum temperature of 900 °C.

The determination of the existence of crystalline phases in the BG powder samples was carried out by X-ray powder diffraction (XRD, Panalytical X’Pert PRO^3^, Almelo, Netherlands) using Cu Kα radiation with a 2θ variation between 5 and 80° and a speed-per-step of 0.026 s^−1^.

The determination of the relevant chemical groups and structural details of the prepared BG samples was performed by FTIR (Mattson Instruments - Galaxy series 7000, Texas City, TX, USA), with the pastilles prepared with potassium bromide (KBr) with a powder-to-KBr weight ratio of 1:150. FTIR analysis was carried out through 256 scans with wavenumber ranging from 4000 to 300 cm^−1^ and a resolution of 4 cm^−1^.

The analysis of the specific surface area (SSA) of the BG powder samples was performed by physical adsorption of N_2_, applying the Brunauer, Emmett, and Teller (BET) method and using micromeritics (Micromeritics Instruments-Gemini 2380 V2.00, Norcross, GA, USA) equipment. The samples had previously been degassed at a temperature of 200 °C.

The morphological features of the BG powder samples were analysed by scanning electron microscope (SEM - Hitachi S-4100, Tokyo, Japan) at 15.0 kV accelerating voltage. For that, it was necessary to fix the samples on an aluminium support using two-sided carbon tape, followed by carbon deposition using a carbon coater (Emitech-K950X Turbo Evaporator, Montigny-le-Bretonneux, France), which creates a thin electron-reflective layer on the surface of the powder particles. In this way, improvement in the conductivity and quality of the images was obtained.

Particle size analysis (PSA) and distribution of the BG powder samples was carried out by laser diffraction using a Coulter LS230 particle size analyzer (Beckman Coulter, Indianapolis, IN, USA).

### 2.4. Evaluation of Antimicrobial Activity

#### 2.4.1. Microbial Strains and Growth Conditions

*Enterococcus faecalis* (ATCC 29212), stored at −80 °C in brain heart infusion broth (BHI broth, PanReac AppliChem, Darmstadt, Germany) with 20% glycerol, and *Candida albicans* (ATCC 11225) stored at the same temperature in Sabouraud dextrose broth (SDB, Liofilchem, Roseto degli Abruzzi, Italy) with 20% glycerol were used in this study.

The bacterium was subcultured in brain heart infusion agar (BHI agar, Liofilchem, Roseto degli Abruzzi, Italy) and, prior to the experiments, one isolated colony was inoculated in 10 mL of brain heart infusion broth and grown anaerobically at 37 °C overnight. *Candida albicans* was subcultured in Sabouraud dextrose agar (SDA, Liofilchem, Roseto degli Abruzzi, Italy) and, before each assay, one isolated colony was inoculated in SDB and grown aerobically at 37 °C overnight with stirring. An aliquot of each culture (300 μL) was transferred into a new fresh liquid medium and grown under the same growth conditions until the stationary growth phase. This culture was then used for the assay.

#### 2.4.2. Growth Inhibition Effect of BG on *E. faecalis* and *C. albicans*

Each microorganism culture, grown overnight and in the stationary phase, was tenfold diluted in culture medium to a final concentration of ~10^8^ colony-forming units per milliliter (CFU mL^−1^) (corresponding to 0.5 Macfarland). The diluted microbial suspension was equally distributed in sterilized microtubes (1 mL per tube) and was incubated with 5 or 15 mg mL^−1^ of each BG sample. A microbial control (comprised of *E. faecalis* and *C. albicans* suspensions) was included. Aliquots of 100 μL of samples and controls were collected at 0, 24, and 48 h of incubation and serially diluted in saline solution. Then, two drops per dilution (10 μL each) were plated on BHI agar for *E. faecalis* and on SDA for *C. albicans* by the drop plate method [36]. The Petri plates were inverted and incubated at 37 °C for 18−24 h. Colonies were counted on the most-appropriate dilution and viable cell concentration was stated as Log_10_ CFU mL^−1^. Three independent experiments, with two replicates each, were performed. The average of the results was calculated.

In order to evaluate the pH variations promoted by the BG samples, the pH of the microbial suspensions incubated with each BG at 5 and 15 mg mL^−1^ and the microbial controls was monitored at 24 h and 48 of incubation using a HALO^®^ wireless pH Meter with micro bulb—HI10832 (Hanna Instruments Inc., Woonsocket, RI, USA). 

### 2.5. Statistical Analysis

The statistical analysis presented was performed using GraphPad Prism 9.0.0 (GraphPad Software, San Diego, CA, USA). The results are presented as the mean of at least 3 independent assays with 3 replicates per assay and the respective mean ± standard deviation. The significance of microbial concentration between BG concentrations plus across the experiments was assessed by two-way analysis of variance (ANOVA). Tukey’s multiple comparison test was used for a pairwise comparison of the means (results presented in Appendix A, Appendix A). The significance of differences was evaluated by comparing the results obtained in the test samples with each other and with the results obtained for the corresponding control samples for the different times. A value of *p* < 0.0001 was considered statistically significant.

## 3. Results

### 3.1. Physical and Morphological Characterization of BG Samples

#### 3.1.1. Thermal Behavior

In order to assess the thermal effects undergone by the synthesized BG samples during calcination, namely their weight variations and the endo- or exothermic nature of the transformations, differential thermal analysis (DTA) and thermogravimetric analysis (TG) were performed simultaneously. The presence of endothermic peaks/bands is usually associated with decomposition reactions that require energy to occur. Oppositely, the presence of exothermic peaks is associated with the occurrence of spontaneous reactions that release energy to the environment, such as the formation of a crystalline phase from an amorphous material, or any combustion reaction. This analysis allowed us to determine the weight loss of the BG samples. Figure 1 shows the DTA/TGA curves of the three BG samples. All of them share common features, including: (i) broad endothermic bands centred at around 200 °C, associated with weight losses that can be attributed to the release of chemically adsorbed water and OH groups from the gel and of hydration water from zinc nitrate (131 °C) and copper nitrate (170 °C); (ii) continuous weight loss within the temperature range from ~330 °C to ~500 °C ascribed to the completion of the processes mentioned in (i) and due to the decomposition of magnesium nitrate (with onset at 330 °C); (iii) relatively sharp endothermic peaks appearing between 550–600 °C associated with noticeable weight loss due to the decomposition of calcium and strontium nitrates with onset temperatures at 561 °C and 570 °C, respectively; (iv) broad exothermic bands of low intensity above ~650 °C, likely due to partial crystallization of the samples, as confirmed below by the XRD results [37].

It is interesting to note that BG1 has undergone a slightly higher total weight loss of ~53% in comparison to BG2 and BG3 (~50%). This difference is attributable to the enrichment of BG2 and BG3 in SrO, whose precursor (Sr(NO₃)₂) being anhydrous, solely contributes to weight loss through the release of nitrate ions during its thermal decomposition.

#### 3.1.2. Crystalline Phases

In order to determine the presence of crystalline phases and their composition, the powder samples of BG were submitted to XRD analysis after calcination at 700 °C. As seen from the XDR patterns (Figure 2), a broad and relatively low intensity peak can be observed in all BG samples due to the presence of a single crystalline phase, identified as calcium strontium silicate (Sr_1.5_Ca_0.5_(SiO_4_)). Interestingly, the XDR peaks tend to become sharper and have increased intensity when moving from BG1 to BG2 and BG3. Considering that the same experimental procedure was adopted in the preparation of all the BG samples, the observed differences can only be attributed to their compositional variations. It is worth mentioning that a simpler quaternary BG with a composition consisting of 60 SiO_2_—34 CaO—4 MgO—2 P_2_O_5_ (mol%) appeared to be completely amorphous after calcination at 700–800 °C when using a mixing period of 60 min over the entirety of the sol composition [32], as in the present work. The lower content of the main glass-network former (SiO_2_) in the present BG compositions is likely to lead to less polymerized and thermally stable glass structures, therefore making them more prone to devitrification when subjected to heat. This helps explain why the investigated BG samples calcined at 700 °C have undergone partial crystallization. Furthermore, their different devitrification extents can be attributed to the various content and types of glass modifiers (CaO, MgO, SrO, CuO). The occurrence of liquid–liquid phase separation (LLPS) is a very common phenomenon in glasses containing different alkaline earth oxides, being ubiquitous to glass in general (e.g., Pyrex borosilicate glass) [38,39]. The presence of LLPS is unavoidable and sometimes desirable, as microscopic heterogeneities could be tuned to produce required properties [40]. The occurrence of partial glass crystallization in the present case may reduce the rate of BG degradation, which could be an advantage when the long-term presence of BG is desirable for the intended application.

#### 3.1.3. Chemical Functional Groups

The FTIR spectra (Figure 3) obtained for the three BG samples within a 250–3750 cm^−1^ spectral range allows identification of the presence of chemical functional groups from 1500 cm^−1^ upwards and the fingerprint region of the material below 1500 cm^−1^. In the FTIR spectra it is possible to observe stretching peaks associated with the functional groups: at ~500 cm^−1^ due to rocking Si-O-Si vibration, at ~910 cm^−1^ due to PO_4_^−3^ symmetric stretching, at ~1000 cm^−1^ due to Si-O-Si stretching, at ~1400 cm^−1^ due to CO_3_^−2^ asymmetric stretching, at ~1630 cm^−1^ due to H-bond to molecular water that was absorbed by the sample, and finally at ~3400 cm^−1^ due to O-H stretching [41,42].

#### 3.1.4. Morphology, Surface Area, and Particle Size

In order to evaluate the existence of pores and their structure in the particle surface of the BG samples, BET analysis was performed by the adsorption of N_2_. The adsorption and desorption isotherms curves (Figure 4) allow classification according to type of isotherm, which is related to pore structure, and according to type of hysteresis loop, which is related to pore characteristics and adsorption mechanism, according to IUPAC. The isothermal curves presented are of type II, associated with microporous materials with heterogeneous surfaces, accompanied by hysteresis loops of type H3, normally associated with type II isotherms, indicating the presence of a network of macro pores that were not completely filled [43].

The morphological features of the BG powder samples after calcination at 700 °C and milling can be observed in the SEM micrographs shown in Figure 5. It can be seen that all samples consist of fine particles with irregular shapes and sizes, heterogeneous appearance, and forming subspherical particle agglomerates. These features are very typical for samples prepared by sol–gel.

Particle size analysis data are reported in Table 2. It is interesting to note that average size tends to decrease from BG1 to BG3, suggesting that the less thermally stable BG3 sample is also less mechanically stable and easier to grind. These results are very consistent and in good agreement with the SEM micrograph observations in Figure 5.

### 3.2. Evaluation of Antimicrobial Activity

#### Growth Inhibition of BG on *E. faecalis* and *C. albicans*

The antimicrobial activity of each BG was evaluated against the two microorganisms primarily responsible for endodontic infections: *E. faecalis* and *C. albicans*. Different BG concentrations were incubated with each microorganism and growth inhibition was evaluated after 24 and 48 h of exposure. The results are presented in Figure 6 and Figure 7. When *E. faecalis* was incubated with BG at the lower concentration (5 mg mL^−1^), no bacterial growth inhibition was observed (ANOVA, *p* < 0.0001) compared to the control, even after 48 h of incubation (Figure 6). However, a different profile was observed when this bacterium was incubated at the higher BG concentration (15 mg mL^−1^) (Figure 6). Although BG1 at 15 mg mL^−1^ did not inhibit growth of *E. faecalis* after 24 or 48 h, the results attained for BG2 and BG3 were remarkable. BG2 at 15 mg mL^−1^ not only inhibited growth but also decreased *E. faecalis* survival by 2.45 Log_10_ (CFU mL^−1^) (ANOVA, *p* < 0.0001) after 48 h of incubation. In the case of BG3 at 15 mg mL^−1^, although no significant effect was achieved in the first 24 h of incubation, a decrease in *E. faecalis* survival of 1.36 Log_10_ (CFU mL^−1^) (ANOVA, *p* < 0.0001) was observed after 48 h.

Regarding *C. albicans*, the effect of BG on growth inhibition and survival was generally more pronounced (Figure 7). Growth inhibition was observed when *C. albicans* was incubated with 5 mg mL^−1^ of BG1–3 for 24 h (ANOVA, *p* < 0.0001), with no recovery after 48 h of incubation. However, at the higher concentration, BG1–3 was shown to inactivate *C. albicans*. For example, BG1 at 15 mg mL^−1^ decreased *C. albicans* survival by 3.45 Log_10_ (CFU mL^−1^) (ANOVA, *p* < 0.0001) in the first 24 h of incubation and 3.85 Log_10_ (CFU mL^−1^) (ANOVA, *p* < 0.0001) after 48 h. A similar profile was achieved for BG2 and BG3, reducing *C. albicans* by ca. 3.97 Log_10_ (CFU mL^−1^) and ca. 3.68 CFU mL^−1^ (ANOVA, *p* < 0.0001), respectively, after 48 h.

Knowing that BGs have the capability of raising the pH of a solution by releasing their cations, which may be related to their antimicrobial activity [3,44], the pH of the microbial suspensions during the antimicrobial activity assays (at 0, 24, and 48 h of incubation) were monitored. The results are presented in Table 3.

The results indicate that immediately after the incubation of the microbial suspension with BG the pH of the solution increases compared to the control. This increase is more significant for the higher BG concentrations (15 mg mL^−1^). However, at the lowest concentration (5 mg mL^−1^), BG1–3 were not able to maintain the elevated pH. For example, in the case of *E. faecalis* and BG1–3 at the lowest concentration, after 24 h of incubation, the pH decreases ca. 7.0, but at 15 mg mL^−1^, the pH remained at ca. 9.0. The pH value increases are more pronounced for *C. albicans*, reaching ca. 10.5 for the highest concentration of BG. In fact, BG1–3 at 15 mg mL^−1^ maintained pH higher than 9.0 after 48 h of incubation.

## 4. Discussion

Interest in exploring the potential antimicrobial properties of bioactive glass has been increasing in recent years. Several mechanisms have been pointed out to explain their mode of action, such as changes in the environmental pH, osmotic pressure, and “needle-like” sharp glass debris that could potentially damage microbial cell walls, creating holes in the cell wall that facilitate the penetration of antimicrobial agents into the cytoplasm [44]. The BGs herein evaluated showed the ability to inhibit the growth of the endodontic pathogens *C. albicans* and *E. faecalis*. Despite the observed alkalinization of the medium, the rise in pH did not exceed the pH tolerated by the microorganisms. Thus, the antimicrobial activity of BGs 1–3 could also be explained by the debris promoted by the materials, which may permeabilize the cell wall to the entrance of the antimicrobial agents present in the BG composition, such as magnesium (Mg), phosphate (P), copper (Cu), zinc (Zn), and strontium (Sr). In fact, in the case of *C. albicans*, BGs at the higher concentration decreased fungal survival more than 3 Log_10_ (CFU mL^−1^). According to the guidelines of the American Society for Microbiology [45], these BGs could be considered antimicrobial agents.

Although it is not possible to correlate the activity of each BG with its composition, our results are in line with the data reported in the literature concerning the antimicrobial activity of BGs with similar compositions. BGs based on SiO_2_—CaO—MgO—P_2_O_5_ (BGs S53P4 and 13–93) and prepared by sol–gel were found to inhibit the growth of a wide selection of clinically important anaerobic pathogens, with particular emphasis on the inhibition of *Bifidobacterium adolescentis* and *Fusobacterium nucleatum* [46]. The same BGs along with four other BGs with similar compositions were screened against 29 clinically important aerobic bacteria [47]. In all materials tested, growth inhibition was demonstrated, although the concentration and time needed for the effect varied depending on the specific BG. The most effective BG, S53P4, had a clear growth-inhibitory effect on all pathogens tested, mainly in the inhibition of *Neisseria meningitidis*, *Yersinia enterocolitica*, and *Enterococcus faecalis,* which is likely due to alkalinization induced by its high sodium content (23%).

The dissolution mechanisms of BG are another parameter that is essential when assessing their antibacterial activity. Zhang et al. [48] reported that the dissolution behavior of six bioactive glasses based on SiO_2_—CaO—MgO—P_2_O_5_ was correlated with their antibacterial effects against 16 clinically important bacterial species, such as *E. coli*, *E. faecalis*, *S. pneumonia*, *S. epidermis*, etc. The results suggested that, in this particular case, the antibacterial effects of BGs are related mainly to the pH increase in the solution and with the concentration of alkaline ions.

BGs composed of SiO_2_—CaO—MgO—P_2_O_5_ and doped with Zn and Sr were tested against *Staphylococcus aureus* and *Escherichia coli* strains [49]. The results showed that the BGs exhibited significant antibacterial efficiency against *S. aureus* and *E. coli.* These materials also preserve good mesenchymal stem cell viability and proliferation. Due to their important biological activity, powdered glasses could find diverse applications in medical areas, namely in dental medicine, for example, as filling material in endodontic applications [50]. In fact, a few studies have pointed out the use BGs in endodontic treatments as alternatives to the gold-standards Ca(OH)_2_ or chlorohexidine. The antimicrobial efficacy exerted by aqueous Ca(OH)_2_ or BG S53P4 powder suspensions on standardized bovine dentin blocks infected with *E. faecalis* was compared in a study by Zehnder et al. [27], who found superior antimicrobial efficacy of BG S53P4 against *E. faecalis* in dentinal tubules when compared to that observed for the Ca(OH)_2_-treated ones. Moreover, while Ca(OH)_2_ was ineffective, the BG suspension eliminated the infection in the sampled dentin layers after 5 days. This work also reported that in a direct exposure test, preincubation with human dentin boosted the BG-killing efficacy against *E. faecalis* ATCC29212, *C. albicans* CCUG19915, *Pseudomonas aeruginosa* ATCC9027, *Streptococcus sanguis* ATCC10556, and *Streptococcus mutans* ATCC25175. The other study that points out the potential use of BGs in endodontic treatments was by Goal et al. [51]. In this clinical study, the effectiveness of chlorhexidine gluconate-1% gel and bioactive glass S53P4 as intracanal medicaments was assessed by using PCR to detect *E. coli* microbial load in the root canal. The results showed that both medicaments considerably reduced bacterial growth, BG S53P4 more-so than chlorhexidine gluconate-1% gel.

As stated, a growing body of evidence has shown that non-doped and metal-doped BGs have higher antimicrobial effectiveness than Ca(OH)_2_ and chlorhexidine, the most used intracanal medicaments. Further, comparative studies regarding the effect of metal doping on BG properties have demonstrated significant improvement of the metal-doped BGs’ antimicrobial activity. Higher efficiency of BGs doped with Zn, Mg, and Sr vs. non-doped BGs against *C. albicans* was recently shown by Ranga et al. [52]. Further, Popa et al. [49] reported *S. aureus* inhibition by silica-based BGs due to the ionic release of Zn and/or Sr antibacterial agents. Stan et al. [53] verified that FastOs^®^BG alkali-free, silica-based BGs with CuO and Ga_2_O_3_ antimicrobial agents reduced *S. aureus* development by four orders of magnitude compared to the control. Taken together, these results support our choice of incorporating different doses of ZnO, CuO, and SrO as antimicrobial agents in BG compositions with a lower degree of glass-network connectivity relative to that reported by Bento et al. [32], to favor the release of the relevant therapeutic, antimicrobial ions. The results presented and discussed support the hypothesis that the combination of different antimicrobial agents in the same BG allows higher antimicrobial activity, perhaps because they possess different inhibition mechanisms and synergistic effects against different microorganisms [49].

## 5. Conclusions

This work focused on sol–gel synthesis of new BGs with different molar concentrations of glass-network modifier oxides, commonly found in bone composition as major constituents (Ca, Mg), but also including other, minor bone constituents endowed with bactericidal activity (Sr, Zn, Cu) and other potential therapeutic effects. The results of their physical, morphological, and biological characterization revealed that all the samples underwent partial crystallization after calcination at 700 °C. Further, the enrichment of BG2 and BG3 with Sr, Zn, and Cu favored devitrification. The particles are porous, as expected for sol–gel-derived bioactive glasses, and can be easily ground to fine powders. Their biological characterization proved that all the BG compositions are effective in inhibiting the growth of two microorganisms with high resistance to intracanal medications, *E. faecalis* and *C. albicans*, which are the main causes of failure of endodontic treatments. The overall results suggest that the synthesized bioactive glass samples are promising for further investigated as alternative intracanal medication.

## Figures and Tables

**Figure 1 nanomaterials-12-01577-f001:**
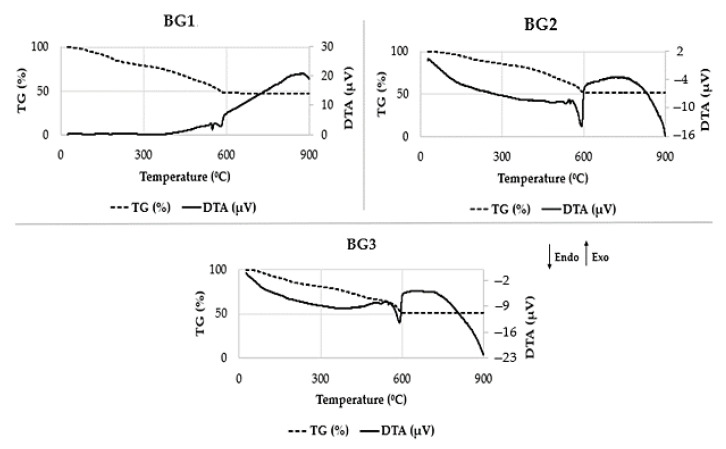
TG/DTA of BG1, BG2, and BG3 powder samples. Initial weight of BG samples: 1 (8.307 mg), 2 (11.391 mg), and 3 (9.353 mg).

**Figure 2 nanomaterials-12-01577-f002:**
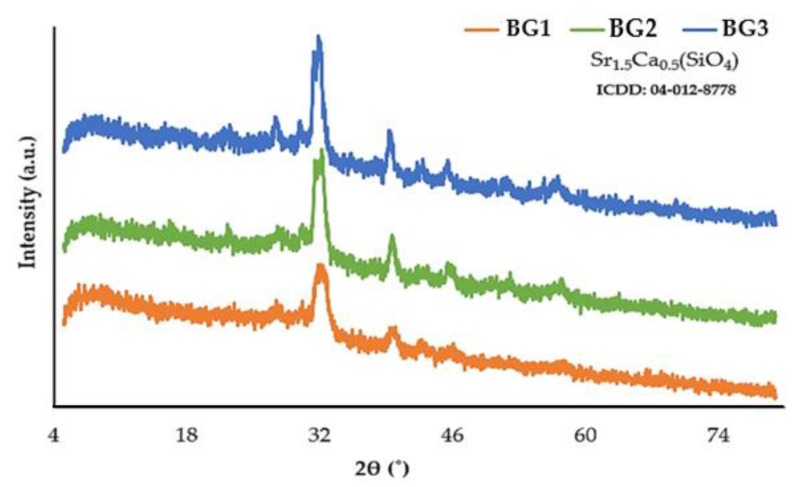
XRD patterns of samples BG1, BG2, and BG3 after calcination at 700 °C. The only crystalline phase identified is calcium strontium silicate (Sr_1.5_Ca_0.5_(SiO_4_)).

**Figure 3 nanomaterials-12-01577-f003:**
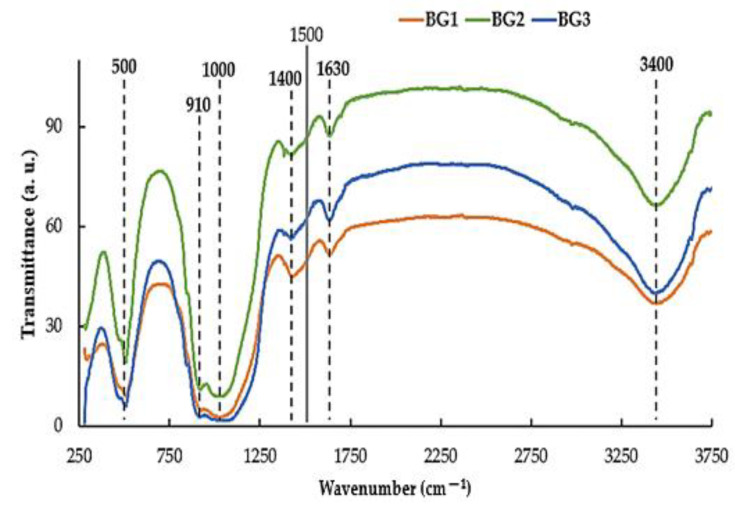
FTIR spectra of samples BG1, BG2, and BG3, after calcination at 700 °C.

**Figure 4 nanomaterials-12-01577-f004:**
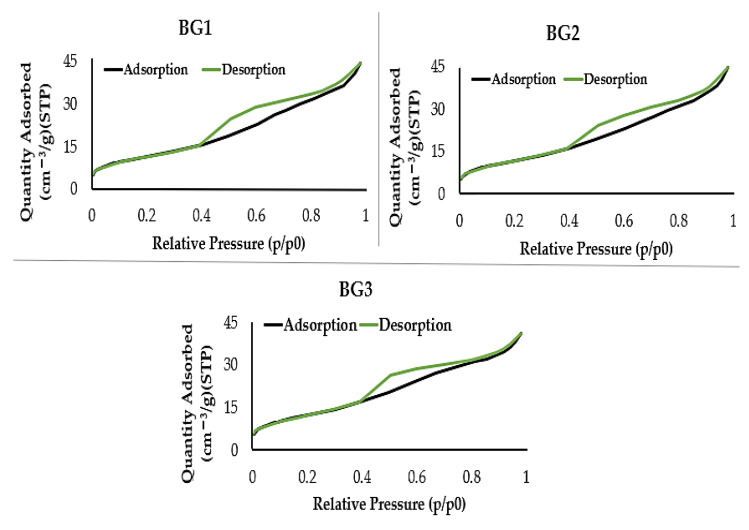
Adsorption and desorption isotherm curves of BG1, BG2, and BG3 powder samples, calcined at 700 °C.

**Figure 5 nanomaterials-12-01577-f005:**
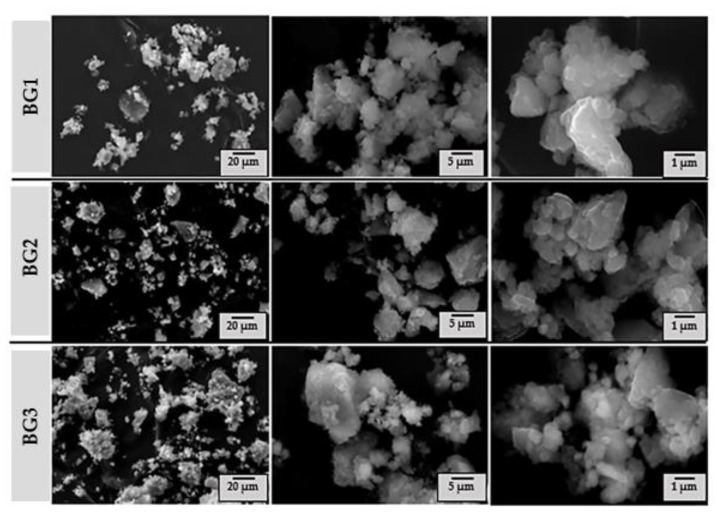
SEM micrographs of the BG powder samples (BG1–3) calcined at 700 °C.

**Figure 6 nanomaterials-12-01577-f006:**
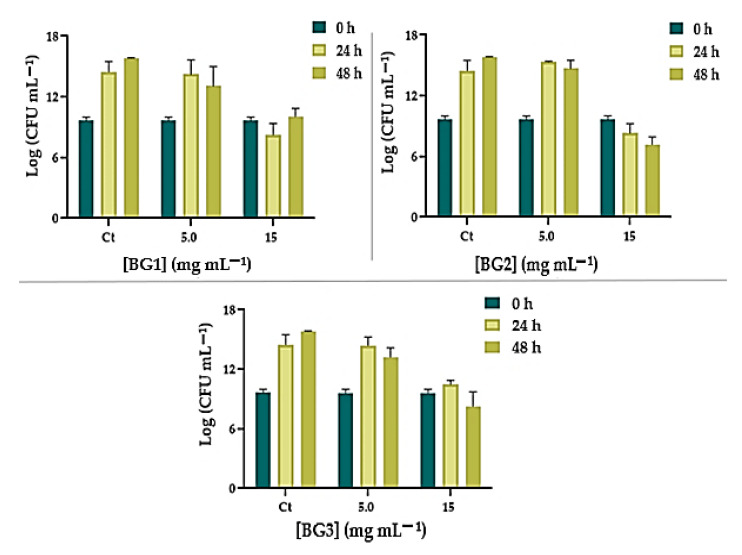
Growth inhibition of *E. faecalis* ATCC 29212 in the presence of 5 and 15 mg mL^−1^ BG1–3 after 24 and 48 h of incubation. The values are expressed as the means of three independent experiments; error bars indicate the standard deviation.

**Figure 7 nanomaterials-12-01577-f007:**
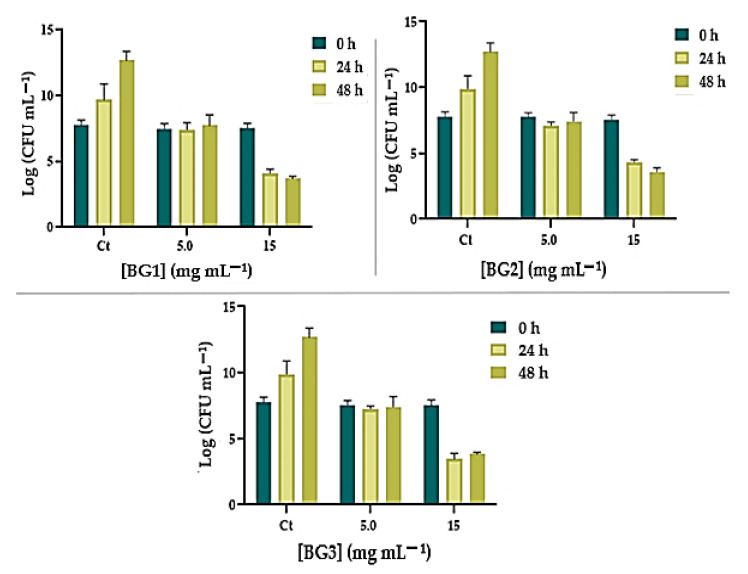
Growth inhibition of *C. albicans* ATCC 11225 in the presence of 5 and 15 mg mL^−1^ of BG1–3 after 24 and 48 h of incubation. The values are expressed as the means of three independent experiments; error bars indicate the standard deviation.

**Table 1 nanomaterials-12-01577-t001:** Chemical compositions and nomenclature of the prepared bioactive glasses.

	Molar %						
Bioactive Glass	SiO_2_	CaO	MgO	SrO	ZnO	CuO	P_2_O_5_
**BG1**	50	20	10	10	6	2	2
**BG2**	50	20	5	15	6	2	2
**BG3**	50	15	5	15	10	3	2

**Table 2 nanomaterials-12-01577-t002:** Relevant particle size distribution data of the BG powder samples (BG1–3).

Bioactive Glass	D_10_ (μm)	D_50_ (μm)	D_90_ (μm)	Average Particle Size (μm)	Standard Deviation (μm)
**BG1**	0.87	2.87	7.86	3.70	2.86
**BG2**	0.85	2.62	5.87	3.03	1.96
**BG3**	0.86	2.36	4.78	2.62	1.52

**Table 3 nanomaterials-12-01577-t003:** pH values of *C. albicans* and *E. faecalis* suspensions incubated with BG1–3 at 5 and 15 mg mL^−1^ at 0, 24, and 48 h.

pH of *Candida albicans* Suspensions					pH of *Enterococcus faecalis* Suspensions				
Samples and Concentrations (mg mL^−1^)		Time 0 h	Time 24 h	Time 48 h	Samples and Concentrations (mg mL^−1^)		Time 0 h	Time 24 h	Time 48 h
**BG1**	**5**	9.53	6.94	6.92	**BG1**	**5**	8.12	7.58	7.67
	**15**	10.20	10.04	9.79		**15**	9.23	9.05	9.06
**BG2**	**5**	9.56	6.88	6.85	**BG2**	**5**	9.03	7.80	7.79
	**15**	10.53	9.93	9.87		**15**	9.13	8.93	8.85
**BG3**	**5**	9.74	6.58	6.89	**BG3**	**5**	8.17	7.36	7.47
	**15**	10.79	10.25	10.21		**15**	9.12	9.06	9.02
*C. albicans* control		5.61	4.56	4.41	*E. faecalis* control		5.34	4.99	5.17
SDB		6.26	6.26	6.26	BHI		7.16	7.16	7.16

## Data Availability

Not applicable.

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
