# Peer review of "New and Efficient Bioactive Glass Compositions for Controlling Endodontic Pathogens"

_nanomaterials, 2022, doi:10.3390/nano12091577_

Round 1
Reviewer 1 Report
Dear Editor,
The article titled “New bioactive glass compositions to foster the success of endodontic treatment” by Bruna L. Correia, Ana T.P.C. Gomes, Rita Noites, José M.F. Ferreira and Ana S. Duarte is well performed experimental study.
I recommend this work for publication after considering the following comments.
1.Page 3.
The formula of tetraethyl orthosilicate (TEOS) is Si(OC2H5)4
and of triethyl phosphate (TEP) is (C2H5)3PO4.
Please make corrections in the text.
2. At the Page 14 it is written that “It is interesting to note that BG1 has undergone a slightly higher total weight loss of ~53% in comparison to BG2 and BG3 (~50%). This difference is attributable to the enrichment of BG2 and BG3 in the anhydrous Sr (NO₃)₂, which does not contribute to weight loss”.
Comment: Although Sr (NO₃)₂ is anhydrous it contributes to weight loss due to its decomposition.
- Page 12.
In conclusions authors have to specify which experimental glass sample (s) are recommended for further investigation.
Yours sincerely,
The reviewer
Reviewer 2 Report
Thanks Author to choose MDPI and Nanomaterials (ISSN 2079-4991)
The manuscript is really interesting and up to date with biblography.
Line 49 value the range of pH.
Line 112 no bond needed for Table 1.
Minor typo present.
Abstract and introduction describe an abundant preamble throughout the manuscript.
Materials and methods are well explained and analyzed.
In discussions and conclusions ask the authors to evaluate these references:
- DOI: 10.3390/ijms22052315
- doi: 10.3390/jfmk5030065
- DOI: 10.1097/SCS.0000000000004734
- DOI: 10.3390/dj8010003
Reviewer 3 Report
The manuscript is interesting, and most of the parts are well written; however, I see few problems that must be corrected.
- In my opinion, the title does not properly point to the content of the article. It is too general, while the research itself has been directed for a specific property.
- In the Introduction, you showed two main areas: technology and antimicrobial properties. After reading the entire work, I think that the work was mainly focused on the development of materials with special properties. Re-build the introduction to clearly show the state-of-the-art in the context of your main aim. You should also pay more attention to experimental materials of this type with antimicrobial properties. These information are partly there, but it requires a slightly better ordering. In my opinion, this part of the text should have more flow. You also did not show in the introduction what exactly the novelty of your work is about; this should be clearly presented in terms of what you have achieved so far, not by "default". What is new in the proposed chemical compositions, and what and how is it supposed to work?
- In the Introduction, the aim and the hypothesis of the work should clearly defined.
- You presented BG1, BG2, and BG3 materials.
- The only properties that you considered in the context of potential ‘clinical success’ were antimicrobial properties. The problem, however, is that it is very difficult to quantify your results in this respect, since there is no control sample in the form of e.g. standard material or something else. So we do not know if the properties you get are typical / better or ‘promising’ (as you wrote in conclusions).
- The results of the post hoc test should be presented (antimicrobial properties). Please combine in table the results of statistical analyses between materials and for each material. It should be compared. Now that is chaotic and hard to judge.
- The discussion is unacceptable, far too short. It requires development and a deeper reflection on the issues discussed.
Other not-important mistakes were noticed:
- Numbering of sections: after 2.4.2 we have 2.7.
Round 2
Reviewer 3 Report
The authors have made significant corrections and the manuscript can be accepted in the future; however, further corrections are needed since not all required changes have been made.
Once again, please show the results of the Tukey post hoc test - thank you for explanations, but they are off-topic because I did not ask for it. You declared in the description of the methodology that Tukey post hoc tests were performed, so it is inconsistent that you did not present their results for microbiological investigations.
I did not ask for a statistical comparison of results with the data from the literature, but for a full presentation of the results of the statistical analysis of your results (ANOVA + Tukey results - now you have only ANOVA). Show this comparison for particular concentrations (influence of time for each material at investigated concentrations), particular times - 24 h and 48 h (influence of concentration for each material) and between BG1, BG2, and BG3 materials (for individual concentrations after 24 h and 48 h).
Moreover, the numbering of the figures in the text has been mixed.
